# Direct Improvement in the Combustion Chamber and the Radiant Surface to Reduce the Emission of Particles in Biomass Cooking Stoves Used in Araucanía, Chile

**Robinson Betancourt Astete** [1,2], **Nicolás Gutiérrez-Cáceres** [1,2,*], **Marcela Muñoz-Catalán** [1,2] **and Tomas Mora-Chandia** [1,2]

1. Mechanical Engineering Department, University of La Frontera, Temuco 4780000, Chile; robinson.betancourt@ufrontera.cl (R.B.A.); marcelanycoll.munoz@ufrontera.cl (M.M.-C.); tomas.mora@ufrontera.cl (T.M.-C.)
2. Center of Waste Management and Bioenergy, University of La Frontera, Temuco 4780000, Chile
* Correspondence: nicolas.gutierrez@ufrontera.cl; Tel.: +569-9857-45787

**Abstract:** Solid particle emissions from burning wood in three internal combustion biomass cooking stoves commonly used in southern Chile were compared. Each stove was used to show differences in sealing systems, combustion chamber shape, and heating surfaces in order to optimize biomass combustion and the energy produced at a low manufacturing cost. The influence of cooking stove design along with particle and gas emissions that resulted from the biomass combustion within the cooking stove was investigated in this study. Levels of diverse atmospheric contaminants, such as particulate matter, emission factor, $NO_x$, $CO_2$, and CO, and the temperature of the flue gases were determined with the Ch-28 method and UNE-EN 12815. The average emission of particulate matter was significantly reduced by modifying the geometry of the combustion chamber and heating surface of each stove, resulting in 5 g/h particle emissions in conventional equipment and 2 g/h in the improved equipment. In relation to gas emissions, there was a 25% maximum decrease in $NO_x$ gases and 35% in CO after modifying the heating surface of each stove. This background supports the evidence of technological improvement with high environmental impact and low economic cost for local manufacturers.

**Keywords:** particle emission; biomass combustion; biomass cooking stoves; domestic heating

## 1. Introduction

Chile is a country that is highly dependent on importing energy, particularly fossil hydrocarbons, even though it possesses a variety of energy resources that are relatively well distributed. Around 24% of the country's power grid comes from forest-based biomass [1], with firewood being the most-used energy source, mainly for heating and cooking purposes: 97% of firewood is used for heating, and the other 3% is used for domestic water heating and, in some cases, for cooking food [2]. The average annual consumption of firewood in a Chilean household depends greatly on the location, due to two fundamental aspects. The first is the geography of the country, where the rainiest and coldest areas are located mainly in the south, starting from the O'Higgins region to the Magallanes region, while the warmest areas are located in the north, starting from the Metropolitan region to the region of Arica y Parinacota. From west to east is the coastal area to the mountain range, where the mountain range zone sees more severe weather than the coastal area, which implies a higher consumption of firewood, encouraged by the abundance of biomass compared to other energy sources such as electricity, fossil fuels, and other sources such as geothermal waters, solar panels, and others. The second is the economic factor, since the country has large socio-economic differences that directly affect access to technologies and fuel for people in the south because the poverty rate is close to 17%, mainly in the Araucanía region [3]. This

poverty rate in the area means that new technologies focused on house improvements, such as thermal insulation and efficient domestic heating, are not feasible for this percentage of the population. According to the latest government reports, fuel poverty in the region has reached 23% and 29% corresponding to the inhabitants without access to electricity supply and domestic hot water [4], which is directly proportional to the socio-economic status of each region; therefore, the few options for the most vulnerable part of the population to acquire biomass stoves are wood-burning stoves, due to its easy installation, versatility, and low price, which is a feasible alternative for this socio-economic sector.

At the national level, wood consumption, depending on the tree species, is obtained mainly from Hualle at 29%, which is equivalent to 3,435,890 $m^3$ st/year, followed by the Eucalyptus globulus with 24%, which is equivalent to 2,872,779 $m^3$ st/year, and finally the remainder corresponds to native and non-native species [5]. In the urban zones of the Aysén region, households consume an average of 18.3 $m^3$ of firewood a year. This goes down to 14.1 $m^3$ in Valdivia, La Union, Paillaco, and Rio Bueno (Los Rios region), 7.7 $m^3$ in Temuco and Padre Las Casas (La Araucanía region), and less in Chillan (Ñuble region) and Rancagua (O'Higgins region) [5]. One cubic meter of firewood is equivalent to approximately 700–900 kg, depending on the type of wood and its water content. Given that firewood is used by thousands of people during the year, wood burning has had severe social and environmental consequences in densely populated cities such as Temuco [2]. These consequences are mainly due to biomass combustion, which is an important source of particulate matter stemming from the incomplete combustion of components like cellulose, hemicellulose, and lignin, in addition to temperature-produced changes caused by combustion from uncontrolled sources [6,7]. According to air quality monitors [8], saturation conditions are present in more than a dozen cities in southern Chile. Saturation conditions occur when the maximum permitted concentration to which the population may be exposed to the environment has been exceeded. In Chile, the Ministry of Health has established air quality standards for coarse particulate matter ($PM_{10}$), D.S. N° 45/2001, and fine particulate matter ($PM_{2.5}$), D.S. N° 12/2011, where the maximum concentration allowed for $PM_{2.5}$ and $PM_{10}$ per year is 20 and 50 $\mu g/m^3$ respectively. The permitted per day concentrations of exposure to PM should be lower than 50 $\mu g/m^3$ on average. Even in the face of the negative implications of firewood, its low cost in comparison to other fuels and the cultural tradition of its use in the country's southern cities make its replacement more difficult for the inhabitants. Implementing measures put in place by ruling governments, such as the atmospheric decontamination plan, the country has attempted to regulate the use of firewood and help create technological initiatives that directly impact the mitigation of contaminants by increasing the efficient and sustainable use of firewood fuels and prioritizing a reduction in atmospheric pollution, such as benefit programs for the replacement of cooking wood stoves, thermal conditioning improvements for housing [9], and to date, the PDTA-100857 project that provides the design and manufacturing so companies can continue improving their cooking wood stoves, on which this study is based.

The objective was to find the best design to improve biomass combustion in common wood-burning stoves in Chile, so that these design enhancements do not increase the manufacturing cost which can affect the final price of the product. Therefore, this design included a combustion chamber and heating surface. We demonstrate how these parameters influence the emission of atmospheric contaminants to help develop efficient, emission-reducing technology, increasing efficiency by 8%.

### 1.1. Biomass as a Fuel

Biomass is defined by the European Standardization Committee as a combination of organic matter derived from vegetable or animal sources or from their natural or artificial transformation, which may undergo energy treatments [10]. Specifically, solid forest biomass of lignocellulose origin is a product of natural and anthropogenic processes. The natural process involves the formation and growth in natural, water-based environments

through photosynthesis, while the artificial process relates to formation through technological production and alterations to the previous natural constituents [11]. There are different types and forms of forest biomass that can be used as energy. On a global scale, 68% of the total bioenergy produced comes from forest biomass [12]. Firewood is cut into logs, ready to be used in domestic fuel apparatuses like stoves. Traditionally, firewood is used in households for cooking and heating. In general, the size of the logs is between 5 and 100 cm [13]. In Chile, the use of forest and agriculture biomass to generate electricity and thermal energy has varied over the last 12 years. On average, it represents 2.6% of the total energy produced [10]. It reached a high of 20% in 2011 and increased by 3% in 2016, coming in third place behind crude oil and carbon. As for firewood and its derivatives, they experienced a 14.7% increase in 2011 over the previous year, while in 2012 it increased 63% (60% residential use and 40% industrial fuel). The use of firewood is distributed between 82.1% in rural and 26.2% in urban homes [10]. Firewood is estimated to be consumed in 1,721,032 homes in Chile, equivalent to 11,926,411 $m^3_{st}$ annually. If the national average of household energy is 10,232 kWh/year, including all fuels and electricity, then firewood represents 46.6% of the fuel used, or 4768 kWh/year. Almost all of its use is dedicated to heating [14]. This, together with the large development of the forestry industry, shows that firewood biomass is an important energy product with good future projections, with increasing demand and comparative advantages over other types of fuel (Figure 1).

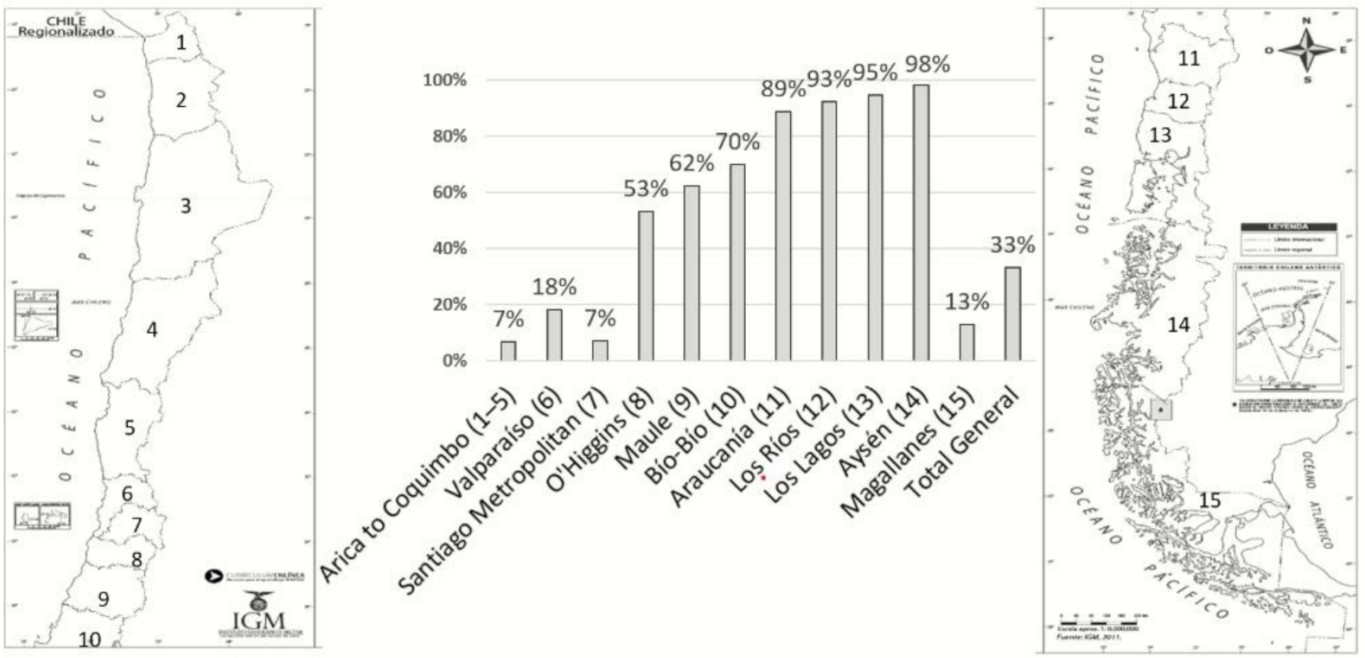

**Figure 1.** Household use of forest biomass in Chile [14].

Figure 1 shows the percentage of firewood consumption by region. Considering the availability of forest resources on a national scale, the highest percentage of firewood consumption and use is in the central-southern zone of the country. This is not surprising, as these are the climatologically colder zones and are located closest to the forest biomass production zones. According to the figures, consumption increases in the fall–winter months (May–August) and is mainly concentrated in the lowest socio-economic sectors [5].

The main wood-burning devices are salamander stoves, open chimneys, simple stoves, double chamber stoves, braziers, and handmade equipment. All of these devices are used, although some are more characteristic in certain areas. The wood fuel in these devices generates fine particulate matter emissions ($PM_{2.5}$), carbon monoxide, volatile components, nitrogen oxides, and other pollutants. As for $CO_2$ emissions, firewood is considered neutral [11].

Of the contaminants mentioned [8], the main problem in Chile is particulate matter ($PM_{2.5}$ and 86 $PM_{10}$), given the large impact on human health [15] that continuous exposure to these particles represents.

### 1.2. Biomass Combustion Process

Combustion of the wood starts when the biomass is exposed to caloric energy. It is followed sequentially by hydrolyzation, oxidation, drying, and pyrolysis, which increases the temperature to form a gaseous fuel. These substances are highly reactive and derived from carbon [16]. The process is as follows: 1. when the wood is exposed to a heating source, its elements start to hydrolyze, dehydrate, and burn as the temperature increases. This process produces volatile fuels, tarry substances, and highly reactive carbonaceous char. 2. When the ignition temperature of these volatiles and chemically treated substances is reached, the combustion process begins. 3. The heat generated by the combustion flame provides the necessary energy for the biomass to gasify and for the flame to spread, further evaporating the water that is found within the cell walls of the biomass, known as the water capillary action. 4. Then, the volatile products (such as water vapor, resinous compounds, and decomposing cell products), the hemicelluloses, and the lignin are separated to then be partially or completely combusted in the flame zone. During combustion, carbon continues to form until the flow of biomass gas falls below the minimum level required to keep the flames. During the flaming combustion, the formation of carbon continues until the volatile fuel flow drops below the minimum level required for the dispersion of the flame. 5. Finally, the smoldering process or the progressive oxidation of the reactive carbon starts. The biomass combustion is characterized by a non-premixed and turbulent flame, given that there is no mixing prior to the combustion reaction between the air and the fuel. In this process, large quantities of particles, which vary in size according to their physical and chemical properties, are emitted into the atmosphere. They can be divided into two categories: non-carbon-based particles, which are generated by non-flammable elements within the fuel, possess neither carbon nor hydrogen atoms [17], and include residual elements that detach from the surface of the fuel [18]. The second category consists of carbon particles formed by pyrolysis of the fuel molecules, and which have not reacted in the flame zone. Other conditions that contribute to the production of solid particles mainly depend on the level of humidity found in the fuel. At some point, moisture in biomass residues significantly affects your ability to heat [19].

## 2. Materials and Methods

This paper investigated three different types of cooking wood stoves commonly used in Chile, corresponding to cooking stove A, B, and C, the descriptions for which are mentioned below. The construction materials are defined in Table 1.

**Table 1.** Construction materials in each device.

| Biomass Cooking Stove | Combustion Chamber | Sealed | Radiant Surface |
|:---:|:---:|:---:|:---:|
| A | Cement | Not sealed | Gray cast iron |
| B | Cement | Glass rope lagging | Gray cast iron |
| C | Cement | Glass rope lagging | Gray cast iron |

Stove A is a conventional model, which can be found in the national market. Stove B is similar to the previous model, except that it has a different cover that allows a hermetic seal, making it more efficient than stove A. The C stove is similar to the B stove except for the combustion chamber, which is delimited by a fin as shown in Figure 2, which reduces the area of the flue gases by 60% over its previous versions. It is considered that only stove A is found for sale in the national market, while the other two are improved prototypes derived from the A device; therefore, those devices are not available in the domestic market yet.

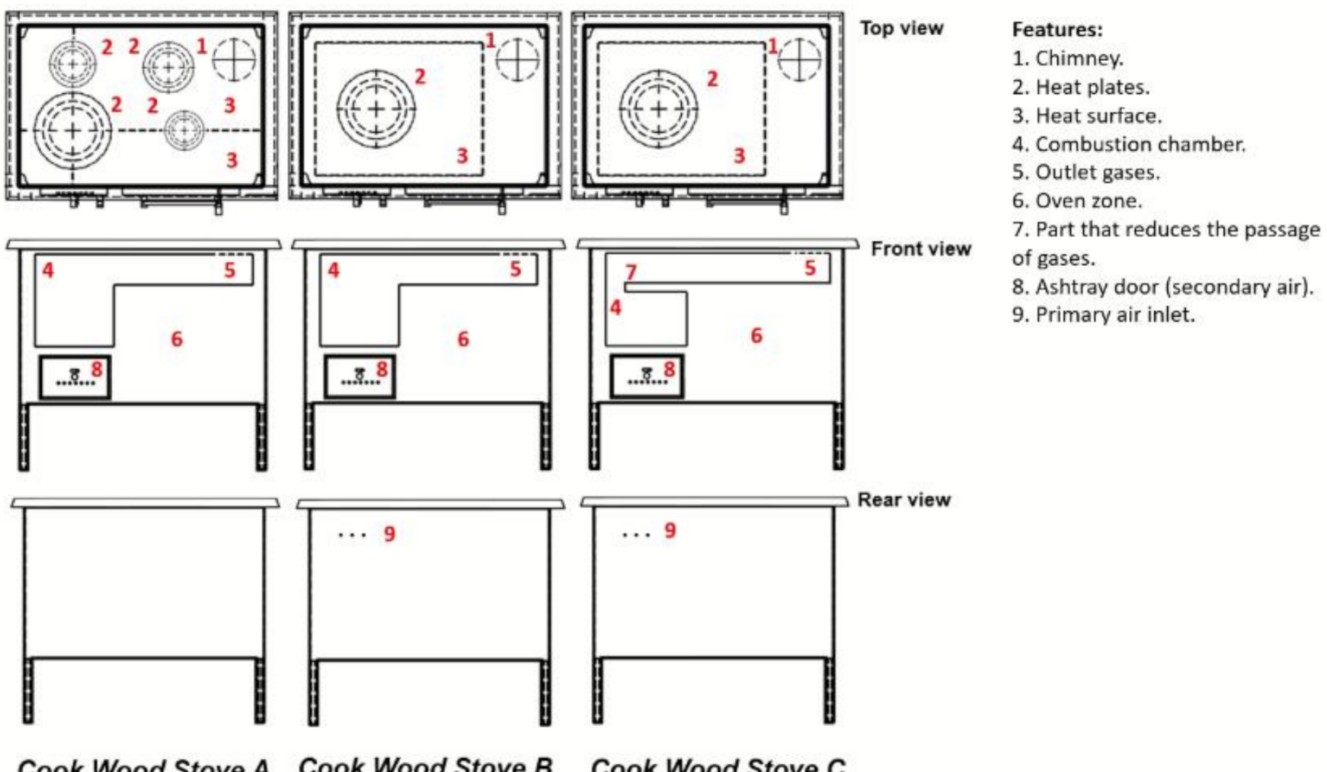

**Figure 2.** Designs of each device used to evaluate the combustion process.

The parameters of the heating surface and combustion chambers of each device are presented in Figure 2. In the case of stove A, the heating surface is divided into six parts: two bigger parts that hold the surface together, and four smaller, circular shapes (heat plates). In stoves B and C, the surfaces are the same, with the surface molded into three bigger parts that secure the surface together with a smaller circular-shaped surface.

The manufacturer has not published the thermal power or efficiency of the devices; hence, one of the main objectives of this research was to determine their performance.

## 2.1. Properties and Improvement in Each Model

The improved stove designs (B and C) were compared to the original cooking stove design A, so the differences between each device are divided into three aspects: air inlets, sealing of the device, and combustion chamber geometry.

### 2.1.1. Air Inlet

The air inlets for devices A, B and C are presented with a yellow arrow, while the red arrow represents parasitic air inlet, in Figure 3.

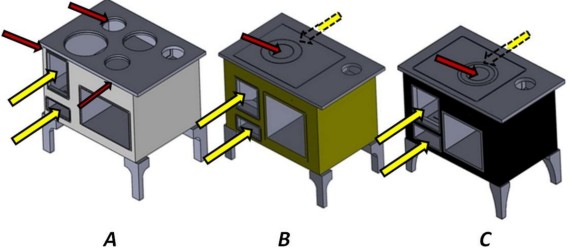

**Figure 3.** The air inlet are pointed by yellow arrows, while the parasite air inlets and unwanted gas outlet are pointed by red arrows, in each stove.

Stove A initially presents two air inlets: the first in the fuel load door and the second in ashtray door. However, the manner of craft manufacture that predominates in these types of devices, which lacks exhaustive reviews and quality inspections, affects the prevalence of manufacturing errors such as fissures between material joints and bindings. As an example, the space between the combustion chamber and the radiant surface allows the parasite air to get in and the unintended combustion gases to get out. Stoves B and C, in this respect, were built in identical ways and have two air inlets; the former has seven holes in the ashtray door, which can regulate its opening through a lever device, while the latter has three holes located in the rear side of the device at the height of the combustion chamber. It is assumed that devices B and C also have parasite air inlets and unwanted gas outlets that were not evaluated in this investigation, shown with red arrows in Figure 3. The air inlets in the ashtray door feed the combustion process in order to keep the fuel burning. The problem is that when the embers and ashes obstruct the air flow on the grate, the combustion process is affected due to the lack of air. To avoid this, three air inlets were added in the rear part of the device (see Figure 2, Feature 9), which maintains the minimal conditions of combustion.

The air flow and movement of combustion gases are seen in Figure 4 and the scheme is the same in each stove, where the passage of the combustion gases through the output duct is regulated by a damper that is entirely closed in all the tests. Consequently, the combustion gases move around the presented section.

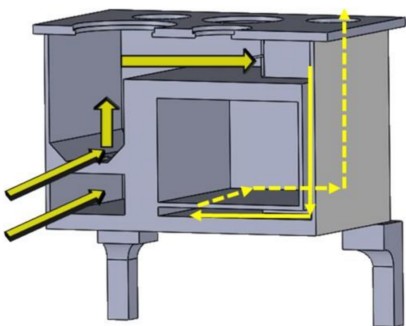

**Figure 4.** Diagram of the flow of air and gases in each stove.

### 2.1.2. Sealing

The optimal sealing for each device is according to the parasite air inlets and the unwanted gas outlets. Considering this, the material used to seal those fissures is fiberglass cord, which can withstand high temperatures without reacting to the flame. Fiberglass cord was used to improve the sealing in the fuel load door and the ashtray door of stoves B and C while the A stove does not have this form of sealing. The sealing also considers the new geometry of the radiant surface: the dilatation and the inherent weight of that surface are enough to create a labyrinth seal to avoid the inlet and outlet of air. The surface's joint type is shown in Figure 5, where the moving part corresponds to the hot plates and the radiant surface (see Figure 2, Features 2 and 3, respectively).

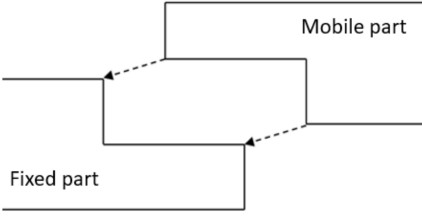

**Figure 5.** Superficial sealing in cooking wood stove B and C.

### 2.1.3. Combustion Chamber Geometry

The chamber combustion design has been limited due to the technical feasibility and the manufacturing that most of the manufacturers in the Araucanía region have. Otherwise, the design changes have been applied only to the C device, whose scheme and differences are presented in Figure 6.

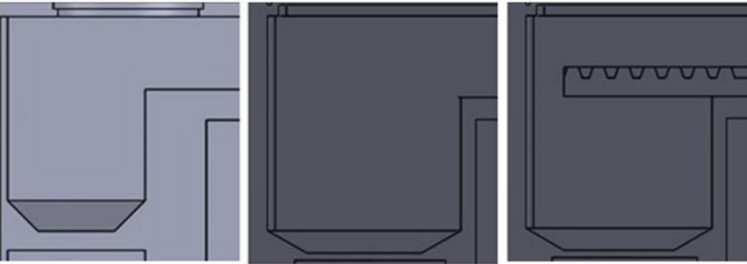

**Figure 6.** Combustion chamber geometry in each stove.

The available combustion chamber volumes in each stove are 0.0343 m$^3$ for stoves A and B, and 0.0214 m$^3$ for stove C, which is much smaller due to the closure made in the combustion chamber that permits a smaller amount of test fuel. From the point of view of dimensions, cooking stoves A and B have similar dimensions in height, width, and length, while cooking stove C has a lower height due to the piece added on the surface of the combustion chamber, reducing its height by 8 cm. The combustion chamber's changes not only consider the geometry and design of the walls, but also the grate from which the ashes descend. Stoves A, B, and C have a conventional grate design. Those grates are presented in Figure 7.

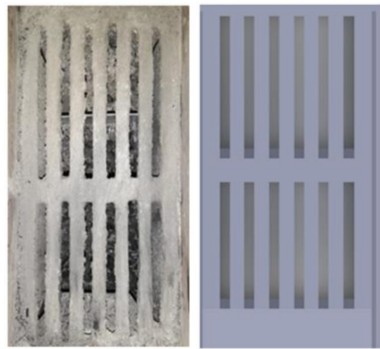

**Figure 7.** Grate geometry, where the ashes descend in each stove.

### 2.2. Properties of Biomass Used

The biomass used was Eucalyptus globulus, which was treated to maintain homogeneity in dimensions, mass, and similar moisture content. Its properties are listed in Table 2. The Chilean method used for this was 5G [20].

**Table 2.** Description of samples.

| Biomass Cooking Stove | Fuel Type | Physical Description |
|:---:|:---:|:---:|
| A | Biomass/sample test wood treated | 310 × 50.8 × 101.6 mm |
| B | Biomass/sample test wood treated | 300 × 50.8 × 100.0 mm |
| C | Biomass/sample test wood treated | 305 × 50.8 × 101.0 mm |

The fuel used in the study was chemically analyzed using two forms of analysis: the first consisted of an elemental and thermogravimetric analysis (Table 3); the second was the fuel load and moisture test (Table 4). The results will be shared next. The chemical properties are expressed in the next table for each sample.

**Table 3.** Thermogravimetric and elemental analysis of Eucalyptus globulus.

| Thermogravimetry of *Eucalyptus globulus* | | |
|---|---|---|
| **Volatile (%)** | Fixed Carbon (%) | Ashes (%) |
| **81.6** | 17.1 | 1.1 |
| Elemental analysis of *Eucalyptus globulus* | | | | | |
| **C (%)** | H (%) | O (%) | N (%) | S (%) | Cl (%) |
| **48.1** | 5.9 | 44.1 | 0.3 | 0.0 | 0.2 |

**Table 4.** Fuel load and moisture (% dry basis) in the test sample.

| Sample | Moisture Content (%) | Std. Dev. | Fuel Load (kg) | Std. Dev. |
|---|---|---|---|---|
| **Biomass for A** | 13.9 | 0.9 | 2.7 | 1.3 |
| **Biomass for B** | 15.0 | 0.9 | 2.9 | 0.3 |
| **Biomass for C** | 15.8 | 1.3 | 2.2 | 0.8 |

### 2.3. Test Method

The experiments were carried out randomly with a previous sampling frame. The methodology used to extract samples from the combustion particulate matter was from the 5G method, and the variables that were set for the use of the cooking wood stove were from studies that mention the predominance of their use, such as the secondary air inlet, which was tested in a single position (lower potential), closed air inlet [21]. The sample was extracted from the chimney or gas evacuation ducts, while the particle extraction was from the dilution duct, given that the hood captures all emissions. Figure 8 shows this process. To extract and sample the combustion gases, a Testo 350 gas analyzer was used, the methodology of which can be found in the NCh3173 and UNE-EN 12815 norms [22]. As for the particle extraction, this was done by collecting samples in 110 mm diameter fiberglass filters made by Pall Corporation. An isokinetic methodology, 5G, was used by the Environmental Supply team to extract the particles. Each stove was used at its lowest potential and tests were repeated six times for each device, giving a total of 12 filters per device and 36 filters across all of the tests. The filters were collected at the end of the total combustion of each device's fuel.

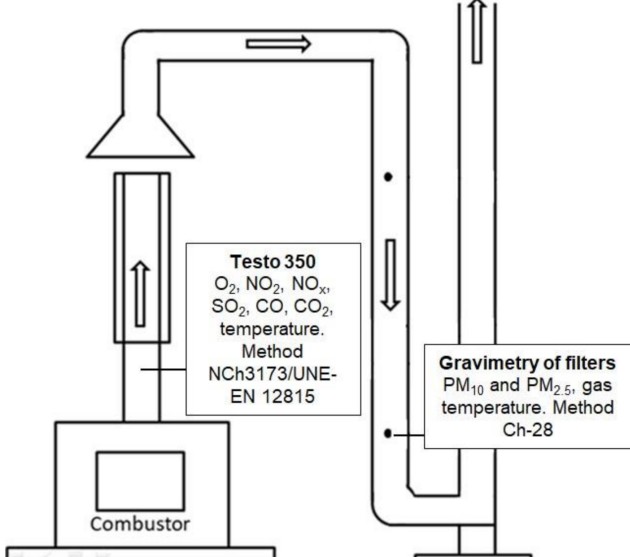

**Figure 8.** Diagram and instrumental design of the gas and particle measurement system.

In relation to the methodology acquisition and monitoring temperature, four Thermocouples Type K temperature sensors of ceramic fiber were used, of which acquisition and monitoring was through the CompactDAQ 9213 and Labview 2011 hardware and software, both from the National Instruments Company. It is worth mentioning that thermocouples are not exposed to direct flame; therefore, correction for radiative temperature is not required. The sampling time was equivalent to a measurement for 1 min, whereas the position of the sensors was the same for each device, located in the radiating surface, combustion chamber, the flame region/area, and in the oven of the different devices, as shown in Figure 9.

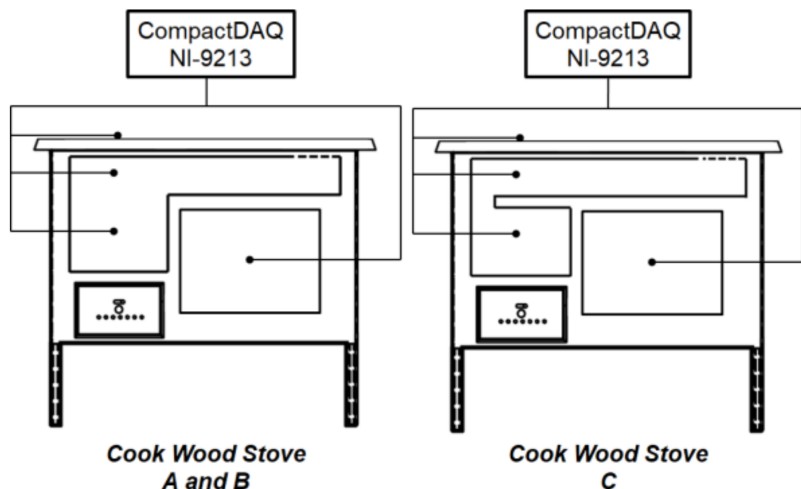

**Figure 9.** Diagram and instrumental design of acquisition and temperature measurement system.

### 2.3.1. Particulate Matter

The calculations of the particulate matter emissions for each device were solved in detail according to the Chilean 5G and Ch-28 methodology and were obtained by the following equation.

$$E = C_s \cdot Q_{std} \tag{1}$$

where $E$ corresponds to the particulate matter emission rate. $C_S$ and $Q_{std}$ are the equivalent to the particulate matter concentrations in the flue gas on a dry basis, which were rectified under standard conditions, and to the average flow rate of the gas flow in the dilution tunnel, respectively. Both values were already calculated using the indicated methods, and it was not the objective of this investigation to explain how the values were obtained. Finally, the emission rate should be applied in the adjustment factor according to the following equation:

$$E_{adj} = 1.82 \cdot E^{0.83} \tag{2}$$

The adjustment factor emission is based on a statistical adjustment that comes from the Monte Carlo simulation defined for population ranges or sample quantities. The details of this factor are presented in the report [23].

### 2.3.2. Performance Measurement

The results of the different combustion processes, referring to the performance evaluation that was carried out for each device, were obtained experimentally through test runs. The combustion efficiency requires the sensitive and the latent heat losses in fumes, in addition to the unburned fuel losses, in which each previously mentioned variable is expressed hereafter.

$$Q_a = (t_a - t_r) \cdot \left[ \frac{C_{pmd} \cdot (C - C_r)}{0.536 \cdot (CO + CO_2)} + \left( C_{pmH_2O} \cdot 1.92 \cdot \frac{(9H + W)}{100} \right) \right] \tag{3}$$

where $t_a$ and $t_r$ are equivalent to the combustion gases and ambient temperature, respectively. $C_{pmd}$ corresponds to the specific heat of dry combustion gases in standard conditions, based on the temperature and gas composition. In addition to that, $C_{pmH_2O}$ corresponds to the water-specific heat in standard conditions depending on the temperature. Moreover, CO and $CO_2$ are the carbon monoxide content and the carbon dioxide content in dry combustion gases, respectively. At last, C, H, and W are the carbon, hydrogen, and the moisture contents in the test fuel.

$$Q_b = 12644 \cdot \left[ \frac{(C - C_r)}{0.536 \cdot 100 \cdot (CO + CO_2)} \right] \tag{4}$$

$C_r$ is equal to the carbon content contained in the residues, regarding the amount of fuel burnt, the approximation for which is given by $C_r = R \cdot b / 100$. In addition, $b$ and $R$ correspond to the fuel components in the residues in relation to the residue material mass and the residue that goes through the grate with regard to the test fuel. In this study, it was measured in each sample.

$$Q_r = 335 \cdot \left[ \frac{R}{100} \right] \tag{5}$$

The combustion efficiency was calculated as:

$$\eta = 100 - \left( \frac{100}{H_u} \cdot (Q_a + Q_b + Q_r) \right) \tag{6}$$

To determine the total thermal power of each device according to the same methodology, the following equation was used:

$$P = \left( \frac{H_u \cdot B \cdot \eta}{100 \cdot 3600} \right) \tag{7}$$

where $B$ is the amount of test mass and $H_u$ is the lower calorific value of the test fuel.

### 2.3.3. Emission Factor

The emission factor represents the number of pollutants emitted into the atmosphere, associated with the activity that generates pollutants [24]. Thus, the emission factor for each device, considering the average emission of particulate matter per fuel quantity, was used for this study, as shown in the following equation.

$$EF = \frac{E_{adj}}{B} \tag{8}$$

## 3. Results

### 3.1. Main Emissions from Combustion

The main emissions of the combustion process correspond to CO, $NO_x$, and $PM_{total}$, which are essential to be able to evaluate the thermal behavior of the appliance and to obtain the respective emission factor or $EF_{PM}$ for each cooking stove, expressed in the summary in Table 5. The reference $O_2$ was 13% in the gas measurement for all samples, and the uncertainty of the gas measurement was less than 5%. The differences of units between concentrations of emission gases and particulate matter were defined by the methods used.

**Table 5.** Summary of emissions and emission factor for each stove.

| | Cooking Stove A | | | Cooking Stove B | | | Cooking Stove C | | |
|---|---|---|---|---|---|---|---|---|---|
| | Sum. Avg. | Std. Dev. | U | Sum. Avg. | Std. Dev. | U | Sum. Avg. | Std. Dev. | U |
| CO (mg/Nm$^3$) | 30.3 | 10.3 | ±2.0 | 34.3 | 16.6 | ±2.0 | 19.7 | 7.6 | ±2.0 |
| NO$_x$ (mg/Nm$^3$) | 4999.2 | 959 | ±10 | 4456.2 | 1293.2 | ±10 | 3748.2 | 260.8 | ±10 |
| PM$_{total}$ (g/h) | 5.4 | 2.7 | ±0.6 | 2.1 | 0.6 | ±0.3 | 4.4 | 1.8 | ±0.24 |
| EF$_{PM}$ (g/kg) | 2.7 | 1.3 | - | 0.9 | 0.2 | - | 2.0 | 0.8 | - |
| | Cooking Stove A | | | Cooking Stove B | | | Cooking Stove C | | |
| | Avg. | Std. Dev. | U | Avg. | Std. Dev. | U | Avg. | Std. Dev. | U |
| CO (mg/Nm$^3$) | 6.081 | 2.115 | ±2.0 | 6.008 | 3.681 | ±2.0 | 5.125 | 3.208 | ±2.0 |
| NO$_x$ (mg/Nm$^3$) | 87 | 35 | ±10 | 63 | 10 | ±10 | 69 | 25 | ±10 |
| PM$_{total}$ (g/h) | 5.4 | 2.7 | ±0.6 | 2.1 | 0.6 | ±0.3 | 4.4 | 1.8 | ±0.24 |
| EF$_{PM}$ (g/kg) | 2.7 | 1.3 | - | 0.9 | 0.2 | - | 2.0 | 0.8 | - |

The CO emissions caused by biomass combustion can be used to indicate the amount of oxygen present in the reaction process. As the fuel evaporates and its mass falls, CO emissions are not reduced. This is because the biomass stove allows air to enter the combustion chamber in uncontrolled proportions, avoiding reaching a thermodynamic state in equilibrium; this situation is apparent in the high variability of emission values in A, and to a lesser extent in B. This situation is reflected in Figure 10, which represents the range of concentration variability emitted by each stove, where C is the one with the lowest emissive trend.

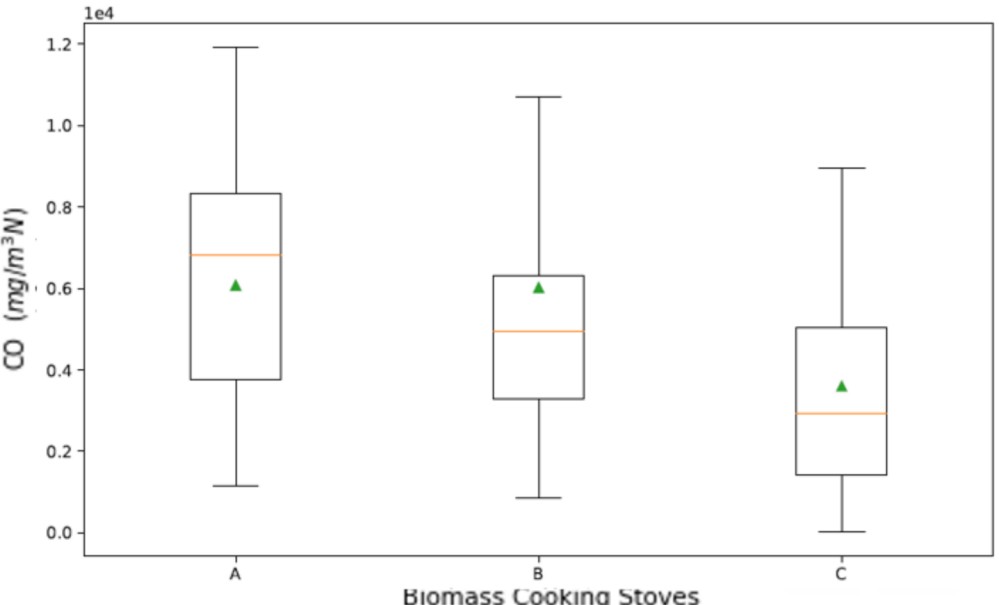

**Figure 10.** Box plot representation of the average CO emission of each stove.

Concentrations emitted per hour of average operation for each cooking stove were 6081, 6008, and 5125 (mg/Nm$^3$), respectively. In comparison with other studies with wood cooking stoves, Koyuncu et al. [25] reported that the emissions of a domestic heating system with similar features were 1489 (mg/MJ), and studies carried out by Boman et al. [26] reported emissions for a different variety of fuel between the ranges of 580 to 2340 (mg/MJ). In this investigation, they were of 1254, 1205, and 1028 (mg/MJ) for CO, the difference of which is between 15 and 30% of what Koyuncu et al. [25] mentioned. This shows evidence of a significant reduction of these emissions. The relationship between the

average concentration of CO emitted and the excess air of the device in each test is also given (Figure 11).

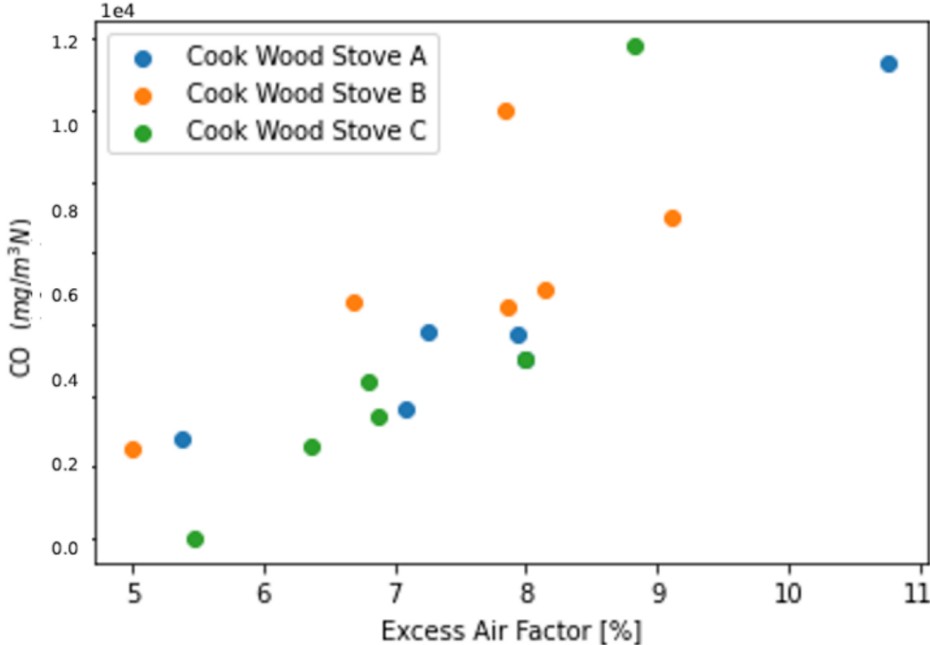

**Figure 11.** Relationship between the average concentration of CO emitted and the excess air measured in each stove.

In Figure 11, the greatest number of points close to each other was obtained with the type C cooking wood stove (green points), while there was a clear dispersion in the results of A and B, meaning that device C exhibited a stable behavior and more efficient combustion due to the relationship between the emission of CO versus the excess air, which means that it used less air for the combustion process and allowed the reduction of emissions of CO. Therefore, it is shown that the emission trend of CO is strongly linked to the air/gas leaks present in the combustion chamber. In summary, the emission behavior from the highest to the lowest was demonstrated by device C, followed by B, and finally by A. Despite the fact that cooking stoves B and C are similar, the throttling at the outlet of the combustion chamber in C (Figure 3) allows combustion to be carried out with less air, which demonstrates the low CO emissions, since the combustion process carried out is more efficient.

The possible gas phase reaction mechanisms for $NO_x$ formation in combustion is described by three mechanisms [27–29]: 1. thermal $NO_x$ is caused by temperatures over 1800 K, which reacts with atmospheric $N_2$ in the combustion chamber; 2. fast $NO_x$ created in the front of the flame; and 3. $NO_x$ is caused by the $N_2$ content in the fuel. In the case of the combustion addressed by this study, it only interacts with the $NO_x$ of the fuel [27,28]. Thus, $NO_x$ emissions obtained in the measurement process represent the formation and reaction of $N_2$ present in the fuel, with a linear correlation between the oxidized $N_2$ and the $NO_x$ emissions [24]. A summary of the $NO_x$ emissions for each cooking stove (Figure 12) is presented.

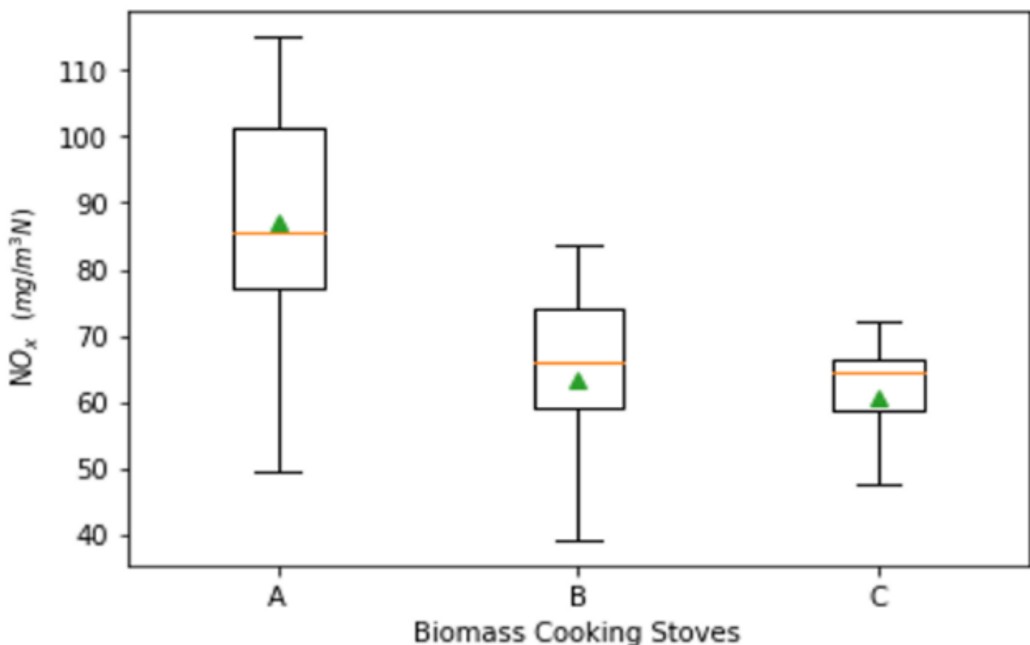

**Figure 12.** Dispersion of the average $NO_x$ emission values in each stove.

The concentrations emitted per hour of average operation of $NO_x$ between stoves A, B, and C were 4999.2, 4456.2, and 3748.2 (mg/Nm$^3$) per hour of sample, respectively. Therefore, a difference or decrease in the emission of $NO_x$ can be ensured. In relation to similar combustion systems, and following the reports presented by Koyuncu et al. [25] where $NO_x$ was 12.54 (mg/MJ), and the results in this case were 10.8, 11.4, and 10.2 (mg/MJ), which differ by 13%, 8%, and 18%, respectively.

As stated previously, the $NO_x$ emissions are associated with the $N_2$ content present in the fuel; however, the differences presented in the assays of cooking stove A in contrast to B and C were substantial. This is due to the non-controlled combustion process itself shown in A, since it has a large number of air inlets, making a stable combustion over time more difficult. This implies the presence of high temperature peaks that can occur inside the flame, causing the formation of thermal $NO_x$, which depends both on the amount of $O_2$ and $N_2$ and on the fuel present in the reaction [30]. The low $NO_x$ emission in C is attributable mainly to the throttle of the gas outlet inside the combustion chamber, which means that the combustion process occurred with a lower amount of air contributing to an efficient combustion.

The particulate matter emissions are also affected by design changes, both in the combustion chamber and in the heating surface. However, the influence of the moisture content of the fuel on the emission of PM is not significant when it is close to $14 \pm 3.6\%$ on a dry basis [31], i.e., there is no statistical support that a variation in humidity like that obtained in the samples significantly affects the emission of particles. A reduction in $PM_{total}$ emission of 62% and 18% was obtained for device B and C, respectively, compared to cooking stove A, as shown in Table 5. Taking other investigations with a different result, Chen et al. [32] reported an $EF_{PM10}$ of $18.1 \pm 6.6$ (g/kg) and $12.7 \pm 1.26$ for $EF_{PM2.5}$. On the other hand, Cooper et al. [33,34] reported that the $EF_{PMtotal}$ for biomass stoves was near to 8.5 (g/kg), demonstrating that the suggested improvements reduce the emission of total particulate matter. It should be considered that stove A does not have seals on its doors, and the cover it uses is, at various points, open to the sample environment, so the results obtained are not accurate, as an unquantified number of particles may be leaving the test environment. Despite the knowledge that this situation can occur, the test was performed in the same way to have a reference, because there is no methodology that combines the measurement of particles emitted by the stove through the chimney and into the testing

environment. The results obtained by B and C presented a much smaller standard deviation than that obtained in A, because the methodology can be applied under the characteristics of B and C, i.e., the results of B and C were valid according to the methodology used. The emission factor was reduced by 66% and 25% for B and C, respectively.

The distribution of total particle emissions for each device is presented in Figure 13. It is evident that stove B was the one with the lowest emission rate compared to the others. This situation is motivated by the type of seal that it has both in the deck and on the door. Stove C, whose characteristics are similar, did not present a concentration lower than B, however, due to the throttling and the area through which the secondary air enters the combustion chamber. Moreover, there is the possibility of producing a second phase of gas oxidation and this generates the second pyrolysis reaction of the PAH, which forms $PM_{Total}$.

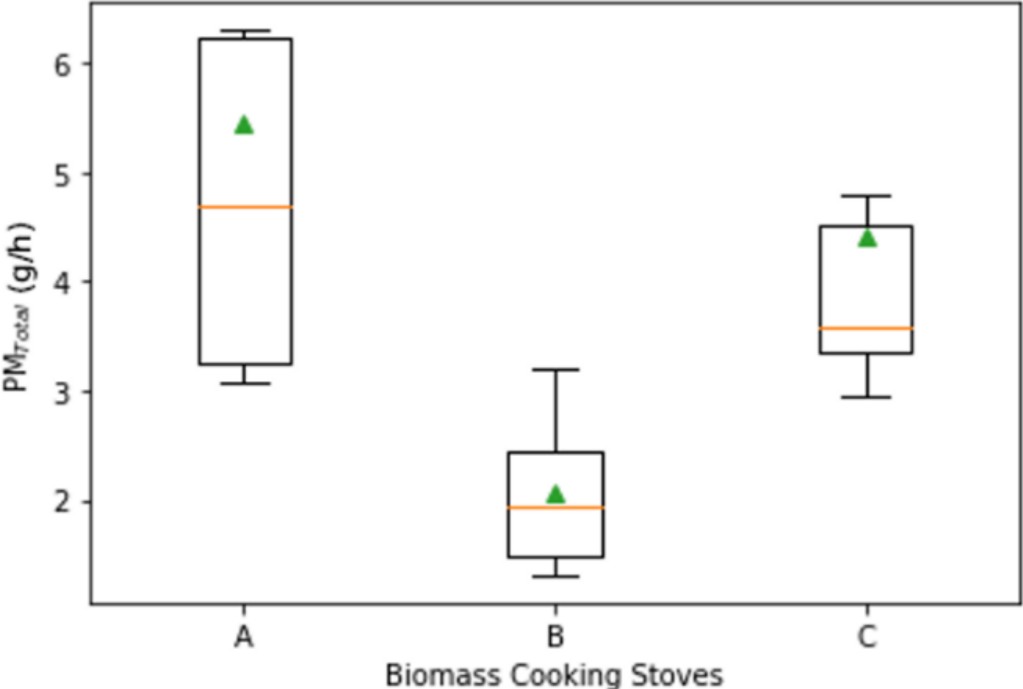

**Figure 13.** Total emission distribution of particulate matter in each stove.

### 3.2. Thermal Behavior of Each Stove

The results of the thermal behavior of the stoves vary depending on the type of surface they have. Devices like wood-burning stoves are designed to raise their surfaces' temperatures, mainly to enable cooking or heating of the surrounding area (see Table 6). The best thermal behavior, then, is found in the device that can maintain constantly high temperatures throughout the sample time. For this test, that device was cooking stove B, which reached a maximum temperature of 192.3 °C, followed by stove C, and, finally, stove A. The high temperatures in stove B can be explained by its combustion chamber, which does not include a system for enclosing flames, so the flames are allowed to pass directly to the stove surface. Similar results were reported by Hueglin et al. [35], in which the thermal behavior of devices that use wood as a fuel were mentioned. Stove C, on the other hand, has a closed combustion chamber that partially retains the flame, as seen in Figure 2. However, these results are not sufficient to define the efficiency of each device.

**Table 6.** Summary of the behavior of the temperature in each biomass cooking stove.

| Cooking Stove | Surface (°C) | | | Combustion Chamber (°C) | | | Exhaust Gases (°C) | | |
|---|---|---|---|---|---|---|---|---|---|
| | Avg. | Std. Dev. | Uncertainty | Avg. | Std. Dev. | Uncertainty | Avg. | Std. Dev. | Uncertainty |
| A | 169.5 | 6.9 | ±8.4 | 420.0 | 75.1 | ±10.5 | 1690.5 | 24.0 | ±4.7 |
| B | 184.9 | 5.0 | ±9.2 | 446.5 | 47.8 | ±11.1 | 154.0 | 10.8 | ±3.8 |
| C | 177.5 | 5.8 | ±8.8 | 578.7 | 63.8 | ±14.4 | 144.6 | 14.4 | ±3.6 |

Combustion gas emissions depend on the combustion process carried out in each device. The resulting temperatures in the combustion chambers are shown in Figure 14. The temperatures in the combustion chambers were noticeably higher in stove C. This can be explained by the shape of the chamber that allows the flame area to have a higher temperature over time. This shows that by carrying out combustion processes in smaller areas and with stable air entrances, it is possible to obtain reactions in elements present in the gases released by the flame [36]. Another variable to determine the thermal behavior in each cooking stove is the temperature of the combustion gas or exhaust temperature, as this temperature is directly related to the emissions and depends on the combustion process carried out in each device. The resulting temperatures in the combustion gas are shown in Figure 14. Exhaust temperature can be an indicator of a stove's thermal efficiency. For stove A, the average temperature was 190 °C. The temperatures for stoves B and C were significantly lower (see Figure 14), with average values of 154 °C and 144 °C, respectively.

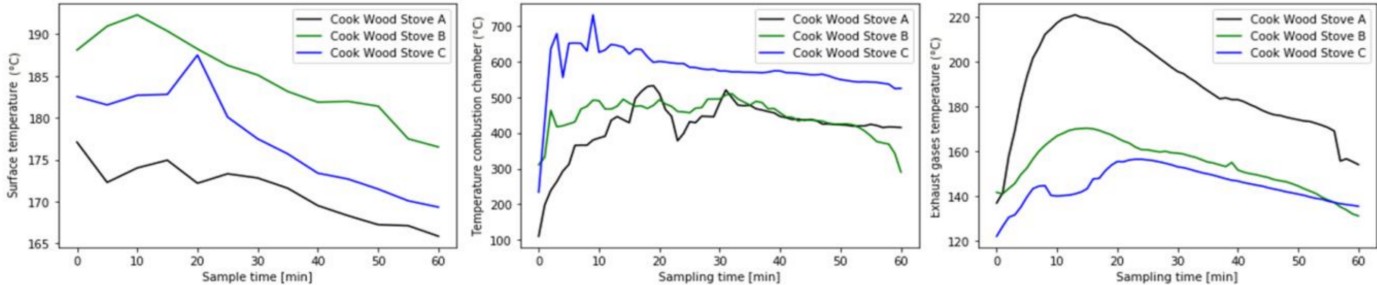

**Figure 14.** Temperature of the heating surface (**left**), combustion chambers (**center**), and exhaust gases temperatures (**right**) during the measurement process for the three stoves.

Studying the temperature of the exhaust gases can highlight two aspects: 1. the time in which the combustion fuels are kept within the combustion space, where reactions of solid and gaseous compounds that do not completely oxidize during the combustion process are favored, which releases energy by radiation to the device's combustion chamber; and 2. the type of heating surface that controls the flame in the stove. High temperatures in the gases and little oxygen can also cause the formation of solid elements. These elements can react to larger particles due to their soot cores [37]. In this study, only point 2 was considered; thus, the stove A showed higher temperatures than stoves B and C, demonstrating lower fuel efficiency, as shown in Table 7. The temperatures found in stoves B and C were similar due to the kind of cover that they have. The shape of stove C had less influence, 7%, on the temperature of the gases than stove B.

**Table 7.** Results of the combustion process in each stove.

| Cooking Stove | Thermal Power (kW) | Std. Dev. | Burning Rate (kg/h) | Std. Dev. | Efficiency (%) | Std. Dev. |
|---|---|---|---|---|---|---|
| A | 6.5 | 1.4 | 1.6 | 0.3 | 73.4 | 1.8 |
| B | 5.6 | 1.2 | 1.4 | 0.2 | 48.7 | 1.5 |
| C | 7.4 | 1.5 | 1.7 | 0.2 | 79.7 | 2.8 |

### 3.3. Performance of Stoves

The results on stove performance and behavior are expressed in Table 7.

The biomass combustion occurring in each stove was significantly different in terms of gas emissions, particulate matter, and also in the temperatures released from the heating surface. Furthermore, the average burning rates were also different for each one, showing a reduction of 12.5% in stove B and an increase of 6.3% in stove C. With the modification of the combustion chamber, the seals of the device and the geometry of the surface increased the efficiency of the devices B and C between 5% and 6%, respectively, as shown in Table 7. The performance, thermal power, and emission of particles of each tested device are expressed in the following graphs with an overview of the values in Figure 15.

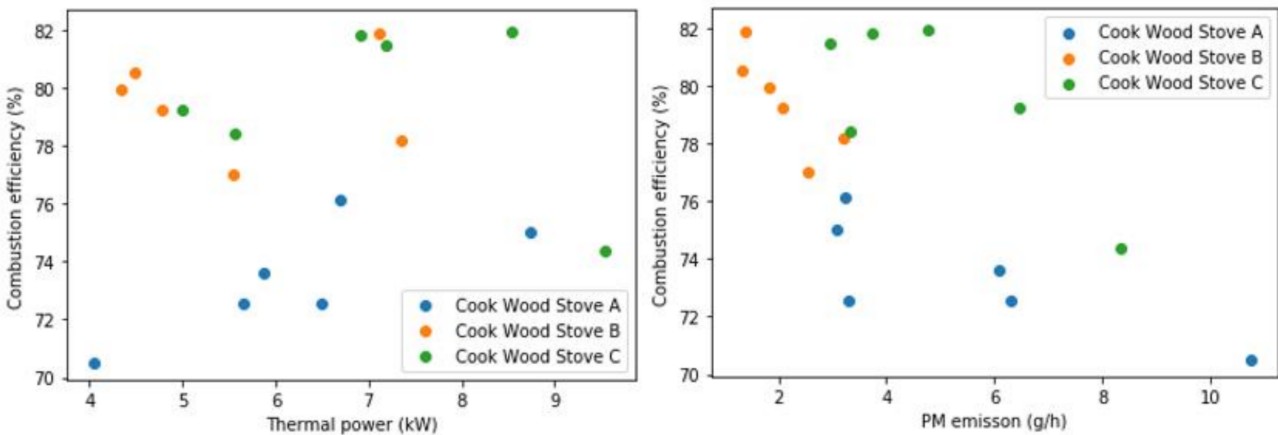

**Figure 15.** Chart combustion efficiency versus thermal power (**left**) and PM emission (**right**) in each stove in all samples.

The relation between combustion efficiency and the thermal power emitted shows that device A is the one that had the lowest relation between efficiency and power, while stove C had the best performance. Their cumulative performance in ascending order was equivalent to 73, 79, and 80% for stoves A, B, and C, respectively. The comparison between the combustion performance and the total cumulative emission of PM shows that stove B performed better than A and C. This comparison showed that higher thermal power and lower combustion performance produce a higher PM emission. The comparison between the combustion performance and the total cumulative PM emission shows that stove B and C performed better than A, which was expected, but through Figure 15, a more stable behavior could be established for stove B, in addition to presenting a smaller dispersion in its results of combustion efficiency versus PM emission. This comparison also showed that higher thermal power and lower combustion efficiency produce a higher PM emission. According to the thermal power generated by each stove and their relation to the burned material rates, stove C is the one that presented the best results since it presented the lowest data dispersion. Even though the power increased with the burning rate, this does not ensure that the process is effective, as shown in the graphs for stove A. The statistical analysis using a Student's *t*-test with a 10% confidence interval provided statistical significance between the samples, which shows that there are changes in the emission factors. However, stove C showed in the graph in Figure 16 that it possessed a lower emission factor to a specific power, whereas stove B behaved similarly to stove C. This was not the case for stove A, which showed greater dispersion data of the emission generated by the power produced.

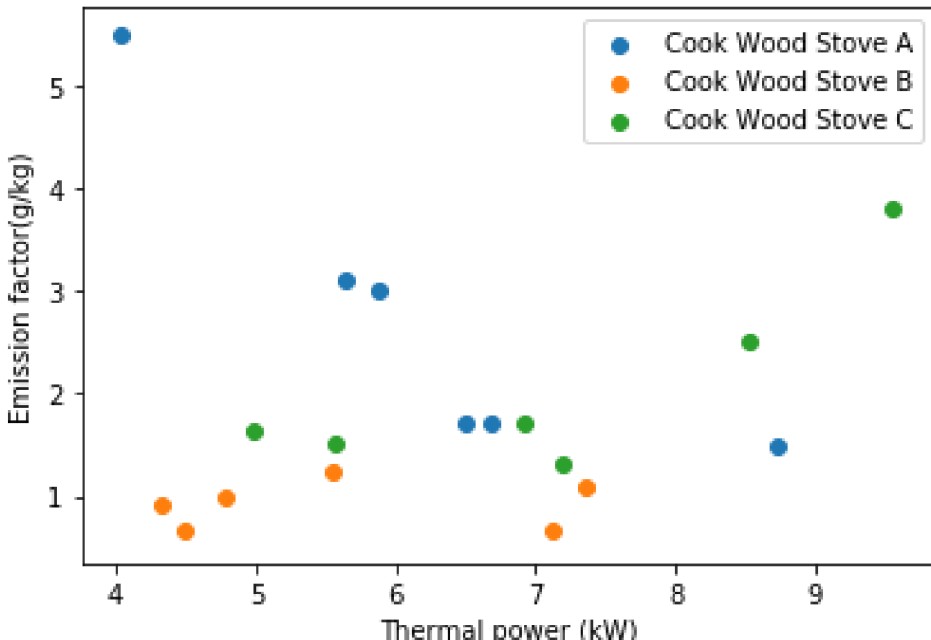

**Figure 16.** Chart emission factor versus thermal power in each stove in all samples.

## 4. Conclusions

The decrease in the formation of particles of versions B and C (compared to the traditional one) was achieved through combustion air control and the adjusted volume of the combustion chamber. This enabled a better use of fuel power, which permitted the reaction of a higher number of reactive volatiles present in the gases, as shown in Figures 11 and 13.

Modifying the heating surface showed that less air was filtered into the interior of the combustion area. This is due to the number of elements, from thirteen to five elements (Figure 2), which comprised the surface and guaranteed a better control of the biomass combustion process, represented in the efficiency obtained (see Table 7).

With the modifications presented in the manuscript, it is possible to reduce the total particulate matter ($PM_{total}$) emissions by 63%, followed by gas emissions, with a maximum reduction of $NO_x$ and CO gases by 25% and 35%, respectively.

Finally, it is concluded that combustion chamber design and heating surfaces favor a biomass combustion process that is much more efficient than the conventional version. This improvement does not suppose an increase greater than 2% of the production cost. Therefore, this technological information has a positive impact on the region, since it is a feasible change that more small and medium-sized manufacturing local companies in Temuco can implement in their designs, which are available to citizens, the regional government, and manufacturers.

**Author Contributions:** In this research work, the contributions of each author are as follows: conceptualization by R.B.A.; methodology and experimentation by N.G.-C. and M.M.-C.; data validated and analyzed by T.M.-C.; research by R.B.A., N.G.-C., and M.M.-C.; R.B.A. contributed material tools and resources; manuscript writing by R.B.A., N.G.-C., and M.M.-C.; review and supervision by T.M.-C. and R.B.A. All authors have read and agreed to the published version of the manuscript.

**Funding:** This research received no external funding.

**Institutional Review Board Statement:** Not applicable.

**Informed Consent Statement:** Not applicable.

**Data Availability Statement:** The information and database for this research are currently not on a platform or website. They can be provided by the corresponding author.

**Acknowledgments:** This research was conducted thanks to the contributions of the Faculty of Engineering and Sciences of Universidad de La Frontera, Temuco, Chile, together with the Laboratory of Combustion and Particles of the Department of Mechanical Engineering and the Center of Waste Management and Bioenergy of the Universidad de La Frontera.

**Conflicts of Interest:** The funders had no role in the design of the study; in the collection, analyses, or interpretation of data; in the writing of the manuscript, or in the decision to publish the results.

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
