# Peer review of "Direct Improvement in the Combustion Chamber and the Radiant Surface to Reduce the Emission of Particles in Biomass Cooking Stoves Used in AraucanÃa, Chile"

_sustainability, doi:10.3390/su13137205_

Round 1
Reviewer 1 Report
The Fig-1 is not clear, should be replaced with high quality image
authors did not measured open burning of wood species as references, so that comparison can be made.
How much particulate mass deposit on ceiling and walls of the chambers
Reviewer 2 Report
The detailed remarks are as follows:
- Authors should avoid „we”, „our”, etc. words in the whole paper. It will be better to change the sentences’ construction in order to use impersonal form. It is more scientific.
- Line 87 „Our objective …” – The objective ….
- Part „Results” – The discussion is poor. More references to the actual state of work and research carried out by other authors should be contained. This point contains only the description of results obtained.
After changes, the paper might be accepted for publication.
Round 2
Reviewer 1 Report
Can be accepted
Reviewer 2 Report
The paper was improved by authors and might be accepted for publication in a present form.
This manuscript is a resubmission of an earlier submission. The following is a list of the peer review reports and author responses from that submission.
Round 1
Reviewer 1 Report
The manuscript describes experiments comparing three wood-fuelled stoves. Emissions and thermal efficiency has been measured. These are indeed important aspects for design of such stoves. However, I don´t see that this manuscript should be published, for the following reasons:
-The authors do no explain what interest this work beyond the design for this specific manufacturer, and I don´t see any interest for a wider audience.
-The three stoves are compared for one setting only. For a plausible comparison between the stoves, a variety of settings (type of wood, amount of wood, air inlet settings, heating water on stove etc.) should be tested.
-The three stoves were fired with fuels of different moisture content (Table 4), so differences between stoves are mixed with differences between fuels.
Moreover, the description and the stove design as well as the experiments and the data are difficult to understand, and need to be improved before any publication. A manuscript with some comments is attached.

Author Response
Kindly thanking the review, we present the correction of each point mentioned by you in the attached document. In addition, here are the responses to your corrections. We should mention that some images have been eliminated and replaced by others that improve the understanding of the manuscript.
Point 1: The authors do no explain what interest this work beyond the design for this specific manufacturer, and I don´t see any interest for a wider audience.
Response 1: The manuscript is developed in order to inform the technological improvement of wood stoves carried out in the region of La Araucanía and thus deliver the designs to local manufacturers. We mention it a few times in the manuscript. However, we have added in the introduction a brief section of the reason that has prompted this research, for this see line 36-59 to contextualize the impact of the project, and lines 88-93 with the objective of the work.
Point 2: The three stoves are compared for one setting only. For a plausible comparison between the stoves, a variety of settings (type of wood, amount of wood, air inlet settings, heating water on stove etc.) should be tested.
Response 2: This configuration of samples was carried out in this way because currently in Chile, the use of this type of technology is focused on a low-income economic sector, which is now mentioned in the manuscript (see line 44), whose purpose is residential heating and cooking of food, where the air inlet configuration is predominantly the one used in the test, for the case of conventional kitchen it is closed air inlet (see line 278). Regarding the type of firewood, this is because in Chile, the legislative work aims to consider Eucalyptus Globulus as a certified wood-energy fuel for use in wood-fired heaters, which is why only the Chilean 5G method is mentioned.
Point 3: The three stoves were fired with fuels of different moisture content (Table 4), so differences between stoves are mixed with differences between fuels.
Response 3: Ensuring the biomass combustion tests with the same moisture content is a real challenge, since the moisture measurement has to be carried out, under the CH-28 protocol, before at least 4 hours before the test and at three different points around it. of the log to be used, which implies differences of up to 4% on a dry basis. Therefore, ensuring that the samples have the same humidity, due to the inhomogeneous size and density of each log used, would be a very complex task. However, we have added in the discussions (see line 398) a comment referenced to the influence of humidity on the emission of Particles.
Point 4: Además, la descripción y el diseño de la estufa, así como los experimentos y los datos, son difíciles de comprender y deben mejorarse antes de cualquier publicación. Se adjunta manuscrito con algunos comentarios.
Response 4: We have added and improved the descriptions of the equipment used (see figure 2), and improved the wording of the sections that you mention in the corrections (see lines 241, 275, 370, 398, 508 and 513. Regarding how to improve the presentation of the data, you could indicate or give examples of how it would be more comfortable and orderly, in order to further improve the presentation. Please, if you could indicate us, we would be grateful to you as an investigative team.
Each correction made by us, plus the comments proposed by the other reviewers are presented in the attached manuscript.
With nothing more to add, and hoping for a good reception and the opportunity to continue sharing views regarding the manuscript, I cordially bid you farewell.

Reviewer 2 Report
The paper presents the influence of different heating parameters (e.g. parameters of equipment) on the emission of particles during the biomass combustion. Generally, the paper is interesting but some aspects of the paper need the improvement before the final acceptance.
The detailed remarks are as follows:
1) line 17-22: Authors mean that firewood is a main energy source in Chile. Do authors have information which kind of wood (e.g. beech wood or oak wood) is applied most often?
2) Line 22-23: Authors write that the use of firewood depends on the location of households. Why are the differences between the households in different parts of Chile? Is it caused by the weather conditions or the living standards of habitants?
3) Line 37: Should be PM instead of MP
4) Line 38: "The Ministry of Health has established air quality standards ..." - it will be worth to mention limited values of coarse particles according to national law.
5) Line 42-43: "the country has attempted to regulate the use of firewood and help create technological initiatives that directly impact the mitigation ..." - give some examples in a bracket
6) Line 46-49: Please add one-two sentences about the results of the research
7) I'm not from Chile and I don't know where regions named I-XII in a diagram are located. It will be readable to add a map of Chile whre the regions will be shown.
8) I suggest to shorten parts of the paper concerning the description of devices tested and the processing of biomass.
9) In conclusions I suggest to add one-two sentences about the further planned research.
Author Response
Kindly thanking the review, we present the correction considered in the conclusions. In addition to presenting the document with the corrections of the other reviewers, to which I attach here. We should mention that some images have been eliminated and replaced by others that improve the understanding of the manuscript.
Point 1: line 17-22: Authors mean that firewood is a main energy source in Chile. Do authors have information which kind of wood (e.g. beech wood or oak wood) is applied most often?
Response 1: This information has been added on line 56, as follows:
At the national level, the wood consumption ,depending on the tree species, is obtained mainly from Hualle, with a 29% which is equivalent to 3.435.890 m3 st/year, followed by the Eucalyptus Globulus with a 24% which is equivalent to 2.872.779 m3 st/year, and finally the remainder corresponds to native and non-native species [5].
Point 2: Line 22-23: Authors write that the use of firewood depends on the location of households. Why are the differences between the households in different parts of Chile? Is it caused by the weather conditions or the living standards of habitants?
Response 2: This information has been added on line 36, as follows:
due to two fundamental aspects: 1. The geography of the country, in where the rainiest and coldest areas are located mainly to the South of Chile, starting from the O’Higgins region to the Magallanes region, while in the opposite side are located the warmest, starting from Metropolitana to Arica y Parinacota regions. From West to East is the coastal area to mountain range area, in where the mountain range zone the weather is more severe than the coastal area, which implies a higher consumption of firewood, encouraged by the biomass abundance in comparison to other energy sources such as the electric light, fossil fuels and others sources as the geothermal waters, solar panels, among others. And 2. The economic factor, since the country has big socio-economic differences that directly affect the access to technologies and fuel for the people in the southern area of the country, because the poverty rate is close to 17%, mainly in the Araucanía region [3]. This poverty rate in the area, implies that the new technologies focused to house improvements, such as thermal insulation and efficient domestic heat are not feasible for this percentage of the population. According to the last governmental reports, the fuel poverty in the region has reached 23% and 29% corresponding to the habitants without access to electricity supply and domestic hot water [4], which is directly proportional to the socio-economic status of each region, therefore the few options for the most vulnerable part of the population to adquire biomass stoves, are the cooking wood stove, due to its easy installation, versatility and low price, which is a feasible alternative for this socio-economic sector.
Point 3: Line 37: Should be PM instead of MP
Response 3: Corrected in manuscript.
Point 4: Line 38: "The Ministry of Health has established air quality standards ..." - it will be worth to mention limited values of coarse particles according to national law.
Response 4:
It has been restructured in the manuscript for better expression (see line 72), as follows.
In Chile, the Ministry of Health has established air quality standards for coarse particle matter (PM 10), D.S. N° 45/2001, and fine particle matter (PM 2.5), D.S. N° 12/2011, where the maximum concentration allowed for PM2.5 and PM10 per year is 20 and 50 ug/m3 respectively. The permitted per day concentrations of exposure to PM should be less than 50ug/m3 on average.
Point 5: Line 42-43: "the country has attempted to regulate the use of firewood and help create technological initiatives that directly impact the mitigation ..." - give some examples in a bracket
Response 5: The following has been added (see line 83):
Implementing measures put in place by ruling governments, such as the atmospheric decontamination plan, the country has attempted to regulate the use of firewood and help create technological initiatives that directly impact the mitigation of contaminants by increasing the efficient and sustainable use of firewood fuels and prioritizing a reduction in atmospheric pollution, such as benefit programs of cooking wood stove replacement, thermal conditioning improvements for housing [9], and until now the PDTA-100857 Project that provides the design and manufacture wherewith the companies are able to keep improving their cooking wood stove, in which this investigation is based on.
Point 6: Line 46-49: Please add one-two sentences about the results of the research
Response 6: The following has been added (see line 93). Therefore, our design included a combustion chamber and heating surface. We demonstrated how these parameters influence the emission of atmospheric contaminants to help develop efficient, emission-reducing technology, increasing the efficiency in 8%
Point 7: I'm not from Chile and I don't know where regions named I-XII in a diagram are located. It will be readable to add a map of Chile whre the regions will be shown.
Response 7:
A new image has been added with a diagram of Chile and its respective regional signage.
Point 8: I suggest to shorten parts of the paper concerning the description of devices tested and the processing of biomass.
Response 8: It is not clear to us what he means by reducing the description of the tested devices and the biomass processing. You could indicate an example or mention the idea that may be in the Methodology section.
Point 9: In conclusions I suggest to add one-two sentences about the further planned research.
Response 9: A new point (see line 513), and the complement of the final conclusion (see line 518) have been added in the conclusions
With the modifications presented in the manuscript, it is possible to reduce the total particulate matter (PMtotal) emissions by 63%, followed by gas emissions, with a maximum reduction of NOx and CO gases of 25% and 35%, respectively.
Finally, we conclude that combustion chamber design and heating surfaces favour a biomass combustion process that is much more efficient than the conventional version. Whose improvement does not suppose an increase greater than 2% of the production cost. Therefore, this technological information has a positive impact on the region, since it is a feasible precedent that more than 26 small and medium-sized manufacturing local companies of Temuco can implement in their designs, which are available to both citizens, the regional government and manufacturers.

Reviewer 3 Report
The authors mentioned decrease in gases like CO and NOx but did not mention the percent decrease in PM level, According to table 5 emission of PM are high in g/h. In biomass combustion PM is main pollutant other than gases so the abstract and conclusion should include that how much reduction is possible in case of PM,
Author Response
Response to Reviewer 3 Comments
Kindly thanking the review, we present the correction considered in the conclusions. In addition to presenting the document with the corrections of the other reviewers, to which I attach here.
Point 1: The authors mentioned decrease in gases like CO and NOx but did not mention the percent decrease in PM level, According to table 5 emission of PM are high in g/h. In biomass combustion PM is main pollutant other than gases so the abstract and conclusion should include that how much reduction is possible in case of PM,

Response 1: A brief summary of the maximum percentage reduction in gas and particulate emissions has been added to the conclusions. It can be presented on lines 516 of the manuscript.

Round 2
Reviewer 1 Report
While improvements have clearly been made in response to the reveiwers´ comments, my overall conclusion remains. Points 1-4 refer to my original points and the authors´ response.
- This response confirms that this work is of local interest and not much interest beyond the local situation.
- This response confirms that there was a poor experimental setup. The fuel load varies a lot between the three stoves. The experimental design does not clearly show that the results are really an effect of the cookstove design. It might as well be an effect of the different amounts of fuel used (Table 4).
- This is unfortunate. This does not explain why the average differs so much between the three stoves, while the STD for each stove is rather small (Table 4).
- Some improvements have been made. However, at least these issues remain:
- What are the red and yellow arrows in this figure?
- Line 344. What O2 reference and why is it assumed to be 13%?
- Table 5: U is not explained.
- Table 5: [g/kg] kg of what?
- Figure 6. It is still not possible to understand this figure. Are you sure that the arrows are drawn correctly?
- Fig 11: What average is shown?
- The text below Fig 12: I cannot see this in the figure. If you want to claim that C has lower emissions you should make a statistic analysis.
- The experimental design should be properly described. It seems that 6 repetitions in each cookstove have been made, but I am not sure (line 287). Where these in random order?
- A revision of the English language is necessary.
Reviewer 2 Report
Authors include remarks indicated by the reviewer.